# Analyzing Federated Learning through an Adversarial Lens

## Abstract

Federated learning distributes model training among a multitude of agents, who, guided by privacy concerns, perform training using their local data but share only model parameter updates, for iterative aggregation at the server. In this work, we explore the threat of *model poisoning* attacks on federated learning initiated by a single, non-colluding malicious agent where the adversarial objective is to cause the model to mis-classify a set of chosen inputs with high confidence. We explore a number of strategies to carry out this attack, starting with simple *boosting* of the malicious agent's update to overcome the effects of other agents' updates. To increase attack stealth, we propose an alternating minimization strategy, which alternately optimizes for the training loss and the adversarial objective. We follow up by using parameter estimation for the benign agents' updates to improve on attack success. Finally, we use a suite of interpretability techniques to generate visual explanations of model decisions for both benign and malicious models, and show that the explanations are nearly visually indistinguishable. Our results indicate that even a highly constrained adversary can carry out model poisoning attacks while simultaneously maintaining stealth, thus highlighting the vulnerability of the federated learning setting and the need to develop effective defense strategies.

## 1 Introduction

Federated learning introduced by McMahan et al. (2017) has recently emerged as a popular implementation of distributed stochastic optimization for large-scale deep neural network training. It is formulated as a multi-round strategy in which the training of a neural network model is distributed between multiple agents. In each round, a random subset of agents, with local data and computational resources, is selected for training. The selected agents perform model training and share only the parameter updates with a centralized parameter server, that facilitates aggregation of the updates. Motivated by privacy concerns, the server is designed to have no visibility into an agents' local data and training process. The aggregation algorithm is agnostic to the data distribution at the agents.

In this work, we exploit this lack of transparency in the agent updates, and explore the possibility of a single malicious agent performing a *model poisoning attack*. The malicious agent's objective is to cause the jointly trained global model to misclassify a set of chosen inputs with high confidence, i.e., it seeks to introduce a *targeted backdoor* in the global model. In each round, the malicious agent generates its update by optimizing for a malicious objective different than the training loss for federated learning. It aims to achieve this by generating its update by directly optimizing for the malicious objective. However, the presence of a multitude of other agents which are simultaneously providing updates makes this challenging. Further, the malicious agent must ensure that its update is undetectable as aberrant.

**Contributions:** To this end, we *propose a sequence of model poisoning attacks*, with the aim of achieving the malicious objective while maintaining attack stealth. For each strategy, we consider both attack strength as well as stealth. We start with malicious update boosting, designed to negate the combined effect of the benign agents, which enables the adversary to achieve its malicious objective with 100% confidence. However, we show that boosted updates can be detected as aberrant using two measures of stealth, accuracy checking on the benign objective and parameter update statistics. Observing that the only parameter updates that need to be boosted are those that con-

tribute to the malicious objective, we design an *alternating minimization* strategy that improves attack stealth. This strategy alternates between training loss minimization and the boosting of updates for the malicious objective and is able to achieve high success rate on both the benign and malicious objectives. In addition, we show that *estimating* the other agents' updates improves attack success rates. Finally, we use a suite of interpretability techniques to generate visual explanations of the decisions made by a global model with and without a targeted backdoor. Interestingly, we observe that the explanations are nearly visually indistinguishable. This establishes the attack stealth along yet another axis of measurement and indicates that backdoors can be inserted without drastic changes in model focus at the input.

**Summary of Empirical Results:** In our experiments, we consider adversaries which only control a single malicious agent and at a given time step, have no visibility into the updates that will be provided by the other agents. We demonstrate that these adversaries can influence the global model to misclassify particular examples with high confidence. We work with both the Fashion-MNIST Xiao et al. (2017) and Adult Census[1], datasets and for settings with both 10 and 100 agents, our attacks are able to ensure the global model misclassifies a particular example in a target class with 100% confidence. Our alternating minimization attack further ensures that the global model converges to the same test set accuracy as the case with no adversaries present. We also show that a simple estimation of the benign agents' updates as being identical over two consecutive rounds aids in improving attack success.

**Related Work:** While data poisoning attacks (Biggio et al., 2012; Rubinstein et al., 2009; Mei & Zhu, 2015; Xiao et al., 2015; Mei & Zhu, 2015; Koh & Liang, 2017; Chen et al., 2017a; Jagielski et al., 2018) have been widely studied, model poisoning attacks are largely unexplored. A number of works on defending against Byzantine adversaries consider a threat model where Byzantine agents send arbitrary gradient updates (Blanchard et al., 2017; Chen et al., 2017b; Mhamdi et al., 2018; Chen et al., 2018; Yin et al., 2018). However, the adversarial goal in these cases is to ensure a distributed implementation of the Stochastic Gradient Descent (SGD) algorithm converges to 'sub-optimal to utterly ineffective models', quoting from Mhamdi et al. (2018). In complete constrast, our goal is to ensure convergence to models that are effective on the test set but misclassify certain examples. In fact, we show that the Byzantine-resilient aggregation mechanism 'Krum' Blanchard et al. (2017) is not resilient to our attack strategies (Appendix C). Concurrent work by Bagdasaryan et al. (2018) considers multiple colluding agents performing poisoning via model replacement at convergence time. In contrast, our goal is to induce targeted misclassification in the global model by a single malicious agent even when it is far from convergence while maintaining its accuracy for most tasks. In fact, we show that updates generated by their strategy fail to achieve either malicious or benign objectives in the settings we consider.

## 2 FEDERATED LEARNING AND MODEL POISONING

In this section, we formulate both the learning paradigm and the threat model that we consider throughout the paper. Operating in the federated learning paradigm, where model weights are shared instead of data, gives rise to the *model poisoning* attacks that we investigate.

### 2.1 FEDERATED LEARNING

The federated learning setup consists of $K$ agents, each with access to data $\mathcal{D}_i$, where $|\mathcal{D}_i| = l_i$. The total number of samples is $\sum_i l_i = l$. Each agent keeps its share of the data (referred to as a *shard*) private, i.e. $\mathcal{D}_i = \{\mathbf{x}_1^i \cdots \mathbf{x}_{l_i}^i\}$ is not shared with the server $S$. The objective of the server is to learn a global parameter vector $\mathbf{w}_G \in \mathbb{R}^n$, where $n$ is the dimensionality of the parameter space. This parameter vector minimizes the loss[2] over $\mathcal{D} = \cup_i \mathcal{D}_i$ and the aim is to generalize well over $\mathcal{D}_{\text{test}}$, the test data. Federated learning is designed to handle non-i.i.d partitioning of training data among the different agents.

At each time step $t$, a random subset of $k$ agents is chosen for aggregation. Every agent $i \in [k]$, *minimizes the empirical loss over its own data shard* $\mathcal{D}_i$, by starting from the global weight vector $\mathbf{w}_G^t$ and running an algorithm such as SGD for $E$ epochs with a batch size of $B$. At the end

---

[1] https://archive.ics.uci.edu/ml/datasets/adult
[2] approximately for non-convex loss functions since global minima cannot be guaranteed

of its run, each agent obtains a local weight vector $\mathbf{w}_i^{t+1}$ and computes its local update $\boldsymbol{\delta}_i^{t+1} = \mathbf{w}_i^{t+1} - \mathbf{w}_G^t$, which is sent back to the server. To obtain the global weight vector $\mathbf{w}_G^{t+1}$ for the next iteration, any aggregation mechanism can be used. Following McMahan et al. (2017), we use synchronous training (i.e., server waits till it has received updates from all the agents selected for the time step) and weighted averaging based aggregation: $\mathbf{w}_G^{t+1} = \mathbf{w}_G^t + \sum_{i \in [k]} \alpha_i \boldsymbol{\delta}_i^{t+1}$, where $\frac{l_i}{l} = \alpha_i$ and $\sum_i \alpha_i = 1$. We also experiment with the Byzantine-resilient aggregation mechanism 'Krum' (Blanchard et al., 2017). Details are in Appendix C.

## 2.2 Threat Model: Model Poisoning

Traditional poisoning attacks deal with a malicious agent who poisons some fraction of the *data* in order to ensure that the learned model satisfies some adversarial goal. We consider instead an agent who poisons the *model updates* it sends back to the server. This attack is a plausible threat in the federated learning setting as the model updates from the agents can (i) directly influence the parameters of the global model via the aggregation algorithm; and (ii) display high variability, due to the non-i.i.d local data at the agents, making it harder to isolate the benign updates from the malicious ones.

**Adversary Model**: We make the following assumptions regarding the adversary: (i) there is exactly one non-colluding, malicious agent with index $m$ (limited effect of malicious updates on the global model); (ii) the data is distributed among the agents in an i.i.d fashion (making it easier to discriminate between benign and possible malicious updates and harder to achieve attack stealth); (iii) the malicious agent has access to a subset of the training data $\mathcal{D}_m$ as well as to auxiliary data $\mathcal{D}_{\text{aux}}$ drawn from the same distribution as the training and test data that are part of its adversarial objective. Our aim is to explore the possibility of a successful model poisoning attack even for a highly constrained adversary.

A malicious agent can have one of two objectives with regard to the loss and/or classification of a data subset at any time step $t$ in the model poisoning setting:
**1. Increase the overall loss**: In this case, the malicious agent wishes to increase the overall loss on a subset $\mathcal{D}_{\text{aux}} = \{\mathbf{x}_i, y_i\}_{i=1}^r$ of the data. The adversarial objective is in this setting is $\mathcal{A}(\mathcal{D}_m, \{\mathbf{x}_i, y_i\}_{i=1}^r, \mathbf{w}_G^t) = \text{argmax}_{\mathbf{w}_G^t} L(\{\mathbf{x}_i, y_i\}_{i=1}^r, \mathbf{w}_G^t)$, where $L(\cdot, \cdot)$ is an appropriately defined loss function. This objective corresponds to the malicious agent attempting to cause *untargeted misclassification*.
**2. Obtain desired classification outcome**: The malicious agent has data samples $\{\mathbf{x}_i\}_{i=1}^r$ with true labels $\{y_i\}_{i=1}^r$ that have to be classified as desired target classes $\{\tau_i\}_{i=1}^r$, implying that the adversarial objective is $\mathcal{A}(\mathcal{D}_m, \{\mathbf{x}_i, \tau_i\}_{i=1}^r, \mathbf{w}_G^t) = \text{argmin}_{\mathbf{w}_G^t} L(\{\mathbf{x}_i, \tau_i\}_{i=1}^r, \mathbf{w}_G^t)$. This corresponds to a *targeted misclassification* attempt by the malicious agent.

In this paper, we will *focus on malicious agents trying to attain the second objective, i.e. targeted misclassification*. At first glance, the problem seems like a simple one for the malicious agent to solve. However, it does not have access to the global parameter vector $\mathbf{w}_G^t$ for the current iteration as is the case in standard poisoning attacks (Muñoz-González et al., 2017; Koh & Liang, 2017) and can only influence it though the weight update $\boldsymbol{\delta}_m^t$ it provides to the server $S$. The simplest formulation of the optimization problem the malicious agent has to solve such that her objective is achieved on the $t^{\text{th}}$ iteration is then

$$\begin{aligned} &\underset{\boldsymbol{\delta}_m^t}{\text{argmin}}\, L(\{\mathbf{x}_i, \tau_i\}_{i=1}^r, \mathbf{w}_G^t), \\ &\text{s.t.} \quad \mathbf{w}_G^t = \mathbf{w}_G^{t-1} + \sum_{i \in [k] \setminus m} \alpha_i \boldsymbol{\delta}_i^t + \alpha_m \boldsymbol{\delta}_m^t. \end{aligned} \tag{1}$$

## 2.3 Experimental setup

In order to illustrate how our attack strategies work with actual data and models, we use two qualitatively different datasets. The first is an image dataset, Fashion-MNIST [3] (Xiao et al., 2017) which consists of $28 \times 28$ grayscale images of clothing and footwear items and has 10 output classes. The

---

[3]Serves as a drop-in replacement for the commonly used MNIST dataset (LeCun et al., 1998), which is not representative of modern computer vision tasks

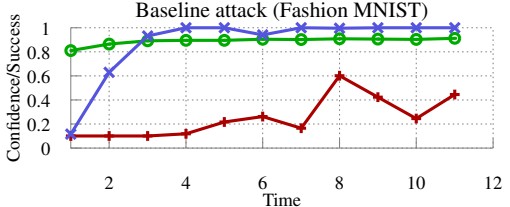 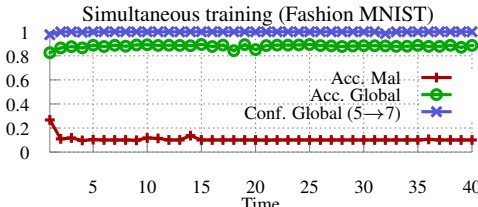

(a) Metrics of interest for baseline (**left**) and simultaneous training attacks (**right**). Unified legend in **right** plot.

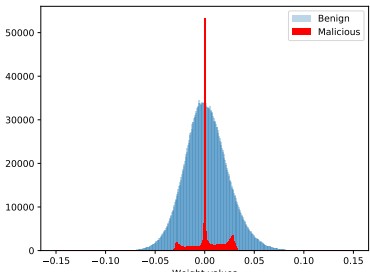 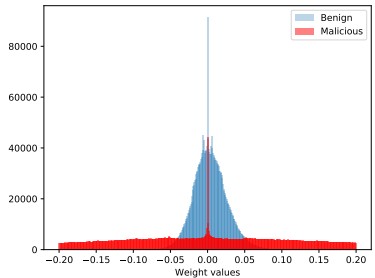

(b) Baseline attack weight update distribution at $t = 4$.

(c) Simultaneous training weight update distribution at $t = 4$.

Figure 1: Metrics of interest and representative weight update distributions for the *baseline* and *simultaneous training* attacks. Figures 1b and 1c show weight update distributions for both benign (**left**) and malicious agents (**right**).

training set contains 60,000 data samples while the test set has 10,000 samples. We use a Convolutional Neural Network achieving 91.7% accuracy on the test set for the model architecture.

The second dataset is the UCI Adult dataset[4], which has over 40,000 samples containing information about adults from the 1994 US Census. The classification problem is to determine if the income for a particular individual is greater (class '0') or less (class '1') than $50,000 a year. For this dataset, we use a fully connected neural network achieving 84.8% accuracy on the test set (Fernández-Delgado et al., 2014) for the model architecture. Owing to space constraints, all results for this dataset are in the Appendix.

For both datasets, we study the case with the number of agents $k$ set to 10 and 100. When $k = 10$, all the agents are chosen at every iteration, while with $k = 100$, a tenth of the agents are chosen at random every iteration. We run federated learning till a pre-specified test accuracy (91% for Fashion MNIST and 84% for the Adult Census data) is reached or the maximum number of time steps have elapsed (40 for $k = 10$ and 50 for $k = 100$). For most of our experiments, we consider the case when $r = 1$, which implies that the malicious agent aims to misclassify a single example in a desired target class. For both datasets, a random sample from the test set is chosen as the example to be misclassified. For the Fashion-MNIST dataset, the sample belongs to class '5' (sandal) with the aim of misclassifying it in class '7' (sneaker) and for the Adult dataset it belongs to class '0' with the aim of misclassifying it in class '1'.

# 3 STRATEGIES FOR MODEL POISONING ATTACKS

We begin by investigating baseline attacks which do not conform to any notion of stealth. We then show how simple detection methods at the server may expose the malicious agent and explore the extent to which modifications to the baseline attack can bypass these methods.

---

[4]https://archive.ics.uci.edu/ml/datasets/adult

### 3.1 LIMITED INFORMATION POISONING OBJECTIVE

In order to solve the exact optimization problem needed to achieve their objective, the malicious agent needs access to the *current value* of the overall parameter vector $\mathbf{w}_G^t$, which is inaccessible. This occurs due to the nature of the federated learning algorithm, where $S$ computes $\mathbf{w}_G^t$ once it has received updates from all agents. In this case, they have to optimize over an *estimate* of the value of $\mathbf{w}_G^t$:

$$
\mathcal{A}(\mathcal{D}_m, \{\mathbf{x}_i, \tau_i\}_{i=1}^r, \hat{\mathbf{w}}_G^t),
$$
$$
\text{s.t.} \quad \hat{\mathbf{w}}_G^t = f(\mathcal{I}_m^t), \tag{2}
$$

where $f(\cdot)$ is an estimator for $\hat{\mathbf{w}}_G^t$ based on all the information $\mathcal{I}_m^t$ available to the adversary. We refer to this as the limited information poisoning objective. The problem of choosing a good estimator is deferred to Section 4 and the strategies discussed in the remainder of this section make the assumption that $\hat{\mathbf{w}}_G^t \approx \mathbf{w}_G^{t-1} + \alpha_m \boldsymbol{\delta}_m^t$. In other words, the malicious agent ignores the effects of other agents. As we shall see, this assumption is often enough to ensure the attack works in practice.

### 3.2 BASELINE ATTACK

Using the approximation that $\hat{\mathbf{w}}_G^t \approx \mathbf{w}_G^{t-1} + \alpha_m \boldsymbol{\delta}_m^t$, the malicious agent just has to meet the adversarial objective $\operatorname{argmin}_{\boldsymbol{\delta}_m^t} L(\{\mathbf{x}_i, \tau_i\}_{i=1}^r, \hat{\mathbf{w}}_G^t)$. Depending on the exact structure of the loss, an appropriate optimizer can be chosen. For our experiments, we will rely on gradient-based optimizers such as SGD which work well for neural networks. In order to overcome the effect of scaling by $\alpha_m$ at the server, the final update $\tilde{\boldsymbol{\delta}}_m^t$ that is returned, has to be *boosted*.

**Explicit Boosting:** Mimicking a benign agent, the malicious agent can run $E_m$ steps of a gradient-based optimizer starting from $\mathbf{w}_G^{t-1}$ to obtain $\tilde{\mathbf{w}}_m^t$ which minimizes the loss over $\{\mathbf{x}_i, \tau_i\}_{i=1}^r$. The malicious agent then obtains an initial update $\tilde{\boldsymbol{\delta}}_m^t = \tilde{\mathbf{w}}_m^t - \mathbf{w}_G^{t-1}$. However, since the malicious agent's update tries to ensure that the model learns labels different from the true labels for the data of its choice ($\mathcal{D}_{\text{aux}}$), it has to overcome the effect of scaling, which would otherwise mostly nullify the desired classification outcomes. This happens because the learning objective for all the other agents is very different from that of the malicious agent, especially in the i.i.d. case. The final weight update sent back by the malicious agent is then $\boldsymbol{\delta}_m^t = \lambda \tilde{\boldsymbol{\delta}}_m^t$, where $\lambda$ is the factor by which the malicious agent *boosts* the initial update. Note that if the assumption $\hat{\mathbf{w}}_G^t \approx \mathbf{w}_G^{t-1} + \alpha_m \boldsymbol{\delta}_m^t$ holds, and $\lambda = \frac{1}{\alpha_m}$, then $\hat{\mathbf{w}}_G^t \approx \mathbf{w}_m^t$, implying that the global weight vector should now satisfy the malicious agent's objective. This method indirectly accounts for the presence of the other agents when using a boosting factor of $\frac{1}{\alpha_m}$.

**Implicit Boosting:** While the loss is a function of a weight vector $\mathbf{w}$, we can use the chain rule to obtain the gradient of the loss with respect to the weight update $\boldsymbol{\delta}$, i.e. $\nabla_{\boldsymbol{\delta}} L = \alpha_m \nabla_{\mathbf{w}} L$. Then, initializing $\boldsymbol{\delta}$ to some appropriate $\boldsymbol{\delta}_{\text{ini}}$, the malicious agent can directly minimize with respect to $\boldsymbol{\delta}$.

**Results:** In the attack with *explicit boosting*, the malicious agent runs $E_m = 5$ steps of the Adam optimizer (Kingma & Ba, 2015) to obtain $\tilde{\boldsymbol{\delta}}_m^t$, and then boosts it by $\frac{1}{\alpha_m} = k$. The results for the case with $k = 10$ are shown in the plot on the *left* in Figure 1a. The attack is clearly successful at causing the global model to classify the chosen example in the target class. In fact, after $t = 3$, the global model is highly confident in its (incorrect) prediction. The baseline attack using *implicit boosting* (Figure 2) is much less successful than the ex-

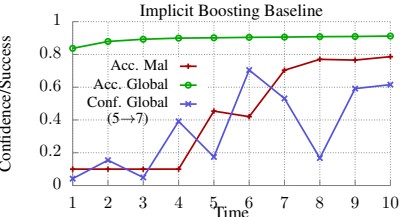

Figure 2: Implicit boosting attack metrics

plicit boosting baseline, with the adversarial objective only being achieved in 4 of 10 iterations. Further, it is computationally more expensive, taking an average of 2000 steps to converge at each time step, which is about $4\times$ longer than a benign agent. Since consistently delayed updates from the malicious agent might lead to it being dropped from the system in practice, we focus on explicit boosting attacks for the remainder of the paper as they do not add as much overhead.

### 3.2.1 MEASURING ATTACK STEALTH AT SERVER

While the baseline attack is successful at meeting the malicious agent's objective, there are methods the server can employ in order to detect if an agent's update is malicious. We now discuss two possible methods and their implication for the baseline attack. We note that neither of these methods are part of the standard federated learning algorithm nor do they constitute a full defense at the server. They are merely metrics that may be utilized in a secure system.

**Accuracy checking:** When any agent sends a weight update to the server, it can check the validation accuracy of $\mathbf{w}_i^t = \mathbf{w}_G^{t-1} + \boldsymbol{\delta}_i^t$, the model obtained by adding that update to the current state of the global model. If the resulting model has a validation accuracy much lower than that of the other agents, the server may be able to detect that model as coming from a malicious agent. This would be particularly effective in the case where the agents have i.i.d. data.

In Figure 1a, the *left* plot shows the accuracy of the malicious model on the validation data (*Acc. Mal*) at each iteration. This is much lower than the accuracy of the global model (*Acc. Global*) and is no better than random for the first few iterations.

**Weight update statistics:** There are both qualitative and quantitative methods the server can apply in order to detect weight updates which are malicious, or at the least, different from a majority of the other agents. We investigate the effectiveness of two such methods. The first, qualitative method, is the visualization of weight update distributions for each agent. Since the adversarial objective function is different from the training loss objective used by all the benign agents, we expect the distribution of weight updates to be very different.

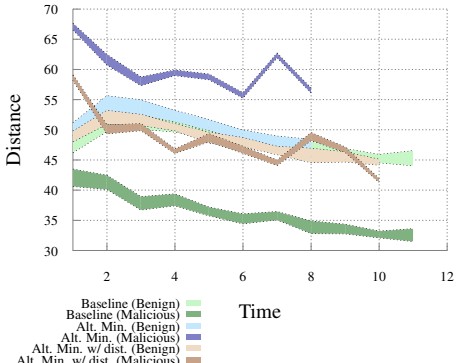

Figure 3: Minimum and maximum $L_2$ distances between weight updates. For each strategy, we show the spread of $L_2$ distances between all the benign agents and between the malicious agent and the benign agents. Going from the baseline attack to the alternating minimization attack with and without distance constraints, we see that the gap in the spread of distances reduces, making the attack stealthier. The benign agents behave almost identically across strategies, indicating that the malicious agent does not interfere much with their training.

This is borne out by the representative weight update distribution at $t = 4$ observed for the baseline attack in Figure 1b. Compared to the weight update from a benign agent, the update from the malicious agent is much sparser and has a smaller range. This difference is more pronounced for later time steps (see Figure 9a in Appendix B).

The second, quantitative method uses the spread of pairwise $L_p$ distances between weight update vectors to identify outliers. At each time step, the server computes the pairwise distances between all the weight updates it receives, and flags those weight updates which are either much closer or much farther away than the others. In Figure 3, the spread of $L_2$ distances between all benign updates and between the malicious update and the benign updates is plotted. For the baseline attack, both the minimum and maximum distance away from any of the benign updates keeps decreasing over time steps, while it remains relatively constant for the other agents. This can enable detection of the malicious agent.

### 3.3 ATTACK WITH SIMULTANEOUS TRAINING

To bypass the two detection methods discussed in the previous section, the malicious agent can try to simultaneously optimize over the adversarial objective and training loss for its local data shard $\mathcal{D}_m$. The resulting objective function is then $\mathrm{argmin}_{\boldsymbol{\delta}_m^t} L(\{\mathbf{x}_i, \tau_i\}_{i=1}^r, \hat{\mathbf{w}}_G^t) + \kappa L(\mathcal{D}_m, \mathbf{w}_m^t)$. Note that for the training loss, the optimization is just performed with respect to $\mathbf{w}_m^t$, as a benign agent would do. When doing explicit boosting, $\hat{\mathbf{w}}_G^t$ is replaced by $\mathbf{w}_m^t$ as well, and the initial weight update $\tilde{\boldsymbol{\delta}}_m^t$ is boosted by $\lambda$ before being sent to the server. This is the only attack strategy explored in concurrent and independent work by Bagdasaryan et al. (2018).

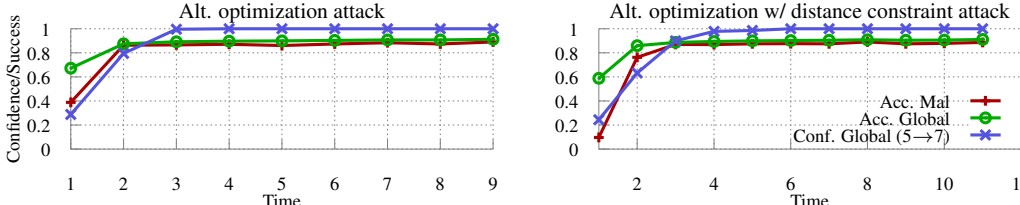

(a) Metrics of interest for alternating minimization attack without (**left**) and with distance constraints(**right**).

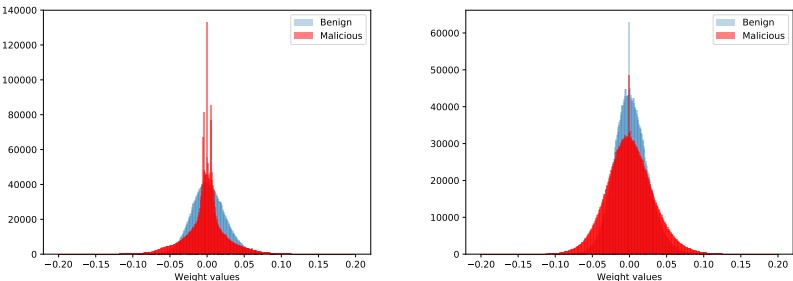

(b) Alternating minimization attack weight update distribution at $t = 4$

(c) Alternating minimization attack with distance constraints weight update distribution at $t = 4$

Figure 4: Metrics of interest and representative weight update distributions for the *alternating minimization* attack with and without distance constraints.

**Results:** In practice, we optimize over batches of $\mathcal{D}_m$ and concatenate each batch with the single instance $\{\mathbf{x}, \tau\}$ to be misclassified, ensuring that the adversarial objective is satisfied. In fact, as seen in Figure 1 in the plot on the *right*, the adversarial objective is satisfied with high confidence from the first time step $t = 1$.

**Effect on stealth:** Since the entire weight update corresponding to both adversarial and training objectives is boosted, the accuracy of $\mathbf{w}_m^t$ on the validation is low throughout the federated learning process. Thus, this attack can easily be detected using the accuracy checking method. Further, while the weight update distribution for this attack (Figure 1c) is visually similar to that of benign agents, its range differs, again enabling detection.

## 3.4 Alternating Minimization formulation

The malicious agent only needs to boost the part of the weight update that corresponds to the adversarial objective. In the baseline attack, in spite of this being the entire update, the resulting distribution is sparse and of low magnitude compared to a benign agent's updates. This indicates that the weights update needed to meet the adversarial objective could be *hidden* in an update that resembled that of a benign agent. However, as we saw in the previous section, boosting the entire weight update when the training loss is included leads to low validation accuracy. Further, the concatenation strategy does not allow for parts of the update corresponding to the two different objectives to be decoupled.

To overcome this, we propose an *alternating minimization* attack strategy which works as follows for iteration $t$. For each epoch $i$, the adversarial objective is first minimized starting from $\mathbf{w}_m^{i-1,t}$, giving an update vector $\tilde{\boldsymbol{\delta}}_m^{i,t}$. This is then boosted by a factor $\lambda$ and added to $\mathbf{w}_m^{i-1,t}$. Finally, the training loss for that epoch is minimized starting from $\tilde{\mathbf{w}}_m^{i,t} = \mathbf{w}_m^{i-1,t} + \lambda \tilde{\boldsymbol{\delta}}_m^{i,t}$, providing the malicious weight vector $\mathbf{w}_m^{i,t}$ for the next epoch. The malicious agent can run this alternating minimization until both the adversarial objective and training loss have sufficiently low values.

**Results:** In Figure 4a, the plot on the *left* shows the evolution of the metrics of interest over iterations. The alternating minimization attack is able to achieve its goals as the accuracy of the malicious

model closely matches that of the global model even as the adversarial objective is met with high confidence for all time steps starting from $t = 3$.

**Effect on stealth:** This attack can bypass the accuracy checking method as the accuracy on test data of the malicious model is close to that of the global model. Qualitatively, the distribution of the malicious weight update (Figure 4b) is much more similar to that of the benign weights as compared to the baseline attack. Further, in Figure 3, we can see that the spread in distances between the malicious updates and benign updates much closer to that between benign agents compared to the baseline attack. Thus, this attack is stealthier than the baseline.

## 3.5 CONSTRAINING THE WEIGHT UPDATE

To increase the attack stealth, the malicious agent can also add a distance-based constraint on $\tilde{\mathbf{w}}_m^{i,t}$, which is the intermediate weight vector generated in the alternating minimization strategy. There could be multiple local minima which lead to low training loss, but the malicious agent needs to send back a weight update that is as close as possible (in an appropriate distance metric) to the update they would have sent had they been benign. So, $\mathbf{w}_m^{i,t}$ is constrained with respect to $\mathbf{w}_{m,ben}^t$, obtained by minimizing the training loss over $\mathcal{D}_m$ starting from $\mathbf{w}_G^{t-1}$, i.e. with the malicious agent mimicking a benign one.

For our experiments, we use the $L_2$ norm as a constraint on $\mathbf{w}_m^{i,t}$, the weight vector obtained at the end of the training loss minimization phase, so $\rho\|\mathbf{w}_{m,ben}^t - \mathbf{w}_m^{i,t}\|_2$ is added to the loss function. Constraints based on the *empirical distribution of weights* such as the Wasserstein or total variation distances may also be used.

**Results and Effect on stealth:** The adversarial objective is achieved at the global model with high confidence starting from time step $t = 2$ and the success of the malicious model on the benign objective closely tracks that of the global model throughout. The weight update distribution for this attack (Figure 4c) is again similar to that of a benign agent. Further, in Figure 3, we can see that the distance spread for this attack closely follows and even overlaps that of benign updates throughout, making it hard to detect using the $L_2$ distance metric.

## 4 IMPROVING ATTACK PERFORMANCE THROUGH ESTIMATION

In this section, we look at how the malicious agent can choose a better estimate for the effect of the other agents' updates at each time step that it is chosen. In the case when the malicious agent is not chosen at every time step, this estimation is made challenging by the fact that it may not have been chosen for many iterations.

### 4.1 ESTIMATION SETUP

The malicious agent's goal is to choose an appropriate estimate for $\boldsymbol{\delta}_{[k]\setminus m}^t = \sum_{i\in[k]\setminus m} \alpha_i \boldsymbol{\delta}_i^t$ from Eq. 1. At a time step $t$ when the malicious agent is chosen, the following information is available to them from the previous time steps they were chosen: i) Global parameter vectors $\mathbf{w}_G^{t_0} \ldots, \mathbf{w}_G^{t-1}$; ii) Malicious weight updates $\boldsymbol{\delta}_m^{t_0} \ldots, \boldsymbol{\delta}_m^t$; and iii) Local training data shard $\mathcal{D}_m$, where $t_0$ is the first time step at which the malicious agent is chosen. Given this information, the malicious agent computes an estimate $\hat{\boldsymbol{\delta}}_{[k]\setminus m}^t$ which it can use to correct for the effect of other agents in two ways:

**Post-optimization correction:** In this method, once the malicious agent computes its weight update $\boldsymbol{\delta}_m^t$, it subtracts $\lambda\hat{\boldsymbol{\delta}}_{[k]\setminus m}^t$ from it before sending it to the server. If $\hat{\boldsymbol{\delta}}_{[k]\setminus m}^t = \boldsymbol{\delta}_{[k]\setminus m}^t$ and $\lambda = \frac{1}{\alpha_m}$, this will negate the effects of the other agents. However, due to estimation inaccuracy and the fact that the optimizer has not accounted for this correction, this method leads to poor empirical performance.

**Pre-optimization correction:** Here, the malicious agent assumes that $\hat{\mathbf{w}}_G^t = \mathbf{w}_G^{t-1} + \hat{\boldsymbol{\delta}}_{[k]\setminus m}^t + \alpha_m \boldsymbol{\delta}_m^{T+1}$. In other words, the malicious agent optimizes for $\boldsymbol{\delta}_m^t$ assuming it has an accurate estimate of the other agents' updates. For attacks which use explicit boosting, this involves starting from $\mathbf{w}_G^{t-1} + \hat{\boldsymbol{\delta}}_{[k]\setminus m}^t$ instead of just $\mathbf{w}_G^{t-1}$.

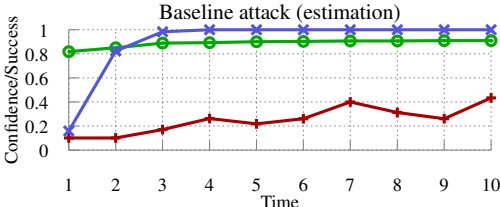 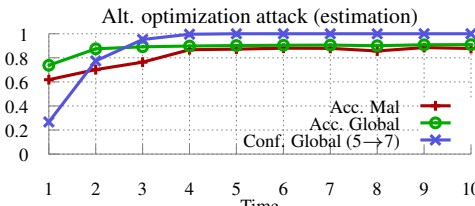

Figure 5: Metrics of interest for the baseline and alternating minimization attacks with explicit boosting and previous step estimation.

## 4.2 ESTIMATION STRATEGIES AND RESULTS

When the malicious agent is chosen at time step $t$ [5], information regarding the probable updates from the other agents can be obtained from the previous time steps at which the malicious agent was chosen.

**Previous step estimate:** In this method, the malicious agent's estimate $\hat{\boldsymbol{\delta}}^t_{[k]\backslash m}$ assumes that the other agents' cumulative updates were the same at each step since $t'$ (the last time step at which at the malicious agent was chosen), i.e. $\hat{\boldsymbol{\delta}}^t_{[k]\backslash m} = \frac{\mathbf{w}^t_G - \mathbf{w}^{t'}_G - \boldsymbol{\delta}^{t'}_m}{t - t'}$. In the case when the malicious agent is chosen at every time step, this reduces to $\hat{\boldsymbol{\delta}}^t_{[k]\backslash m} = \boldsymbol{\delta}^{t-1}_{[k]\backslash m}$.

**Results:** Attacks using previous step estimation with the pre-optimization correction are more effective at achieving the adversarial objective for both the baseline and alternating minimization attacks. In Figure 5, the global model misclassifies the desired sample with a higher confidence for both the baseline and alternating minimization attacks at $t = 2$.

## 5 INTERPRETING POISONED MODELS

Neural networks are often treated as black boxes with little transparency into their internal representation or understanding of the underlying basis for their decisions. Interpretability techniques are designed to alleviate these problems by analyzing various aspects of the network. These include (i) identifying the relevant features in the input pixel space for a particular decision via Layerwise Relevance Propagation (LRP) techniques (Montavon et al. (2015)); (ii) visualizing the association between neuron activations and image features (Guided Backprop (Springenberg et al. (2014)), De-ConvNet (Zeiler & Fergus (2014))); (iii) using gradients for attributing prediction scores to input features (e.g., Integrated Gradients (Sundararajan et al. (2017)), or generating sensitivity and saliency maps (SmoothGrad (Smilkov et al. (2017)), Gradient Saliency Maps (Simonyan et al. (2013))) and so on. The semantic relevance of the generated visualization, relative to the input, is then used to explain the model decision.

These interpretability techniques, in many ways, provide insights into the internal feature representations and working of a neural network. Therefore, we used a suite of these techniques to try and discriminate between the behavior of a benign global model and one that has been trained to satisfy the adversarial objective of misclassifying a single example. Figure 6 compares the output of the various techniques for both the benign and malicious models on a random auxiliary data sample. Targeted perturbation of the model parameters coupled with tightly bounded noise ensures that the internal representations, and relevant input features used by the two models, for the same input, are almost visually imperceptible. This reinforces the stealth achieved by our attacks along with respect to another measure of stealth, namely various interpretability-based detection techniques.

---

[5]If they are chosen at $t = 0$ or $t$ is the first time they are chosen, there is no information available regarding the other agents' updates

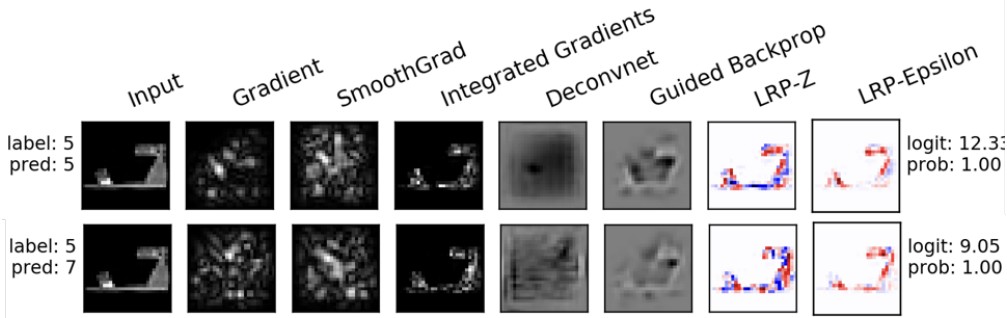

Figure 6: Interpretation of benign (5 → 5) and malicious (5 → 7) model decisions via visualization of feature relevance and representations for a randomly chosen auxiliary data sample.

## 6 DISCUSSION

In this paper, we have started an exploration of the vulnerability of multi-party machine learning algorithms such as federated learning to *model poisoning* adversaries, who can take advantage of the very privacy these models are designed to provide. In future work, we plan to explore more sophisticated detection strategies at the server, which can provide guarantees against the type of attacker we have considered here. In particular, notions of distances between weight distributions are promising defensive tools. Our attacks in this paper demonstrate that federated learning in its basic form is very vulnerable to model poisoning adversaries, as are recently proposed Byzantine resilient aggregation mechanisms. While detection mechanisms can make these attacks more challenging, they can be overcome, demonstrating that multi-party machine learning algorithms robust to attackers of the type considered here must be developed.

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

## A  FURTHER RESULTS

### A.1  RESULTS ON ADULT CENSUS DATASET

Results for the 4 different attack strategies on the Adult Census dataset (Figure 7) confirm the broad conclusions we derived from the Fashion MNIST data. The baseline attack is able to induce high confidence targeted misclassification for a random test example but affects performance on the benign objective, which drops from 84.8% in the benign case to just around 80%. The alternating minimization attack is able to ensure misclassification with a confidence of around 0.7 while maintaining 84% accuracy on the benign objective.

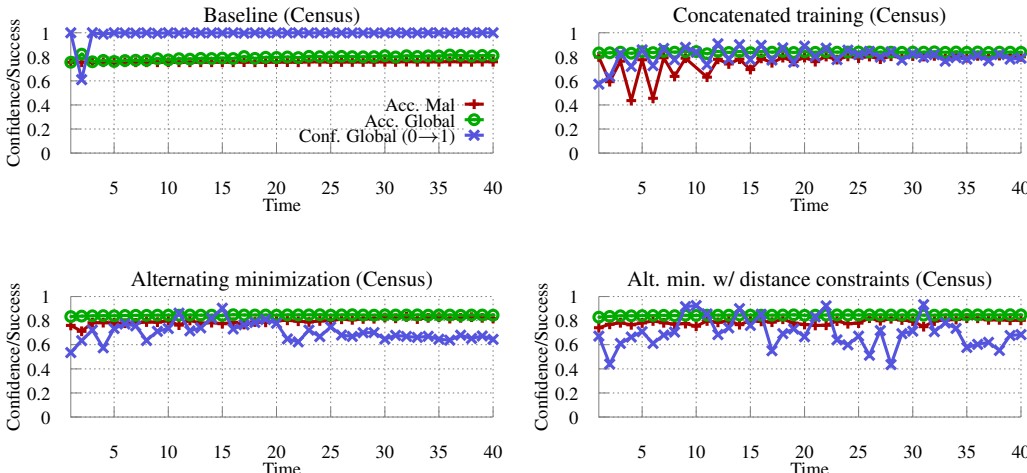

Figure 7: Metrics of interest for 4 different attack strategies with the Adult Census dataset.

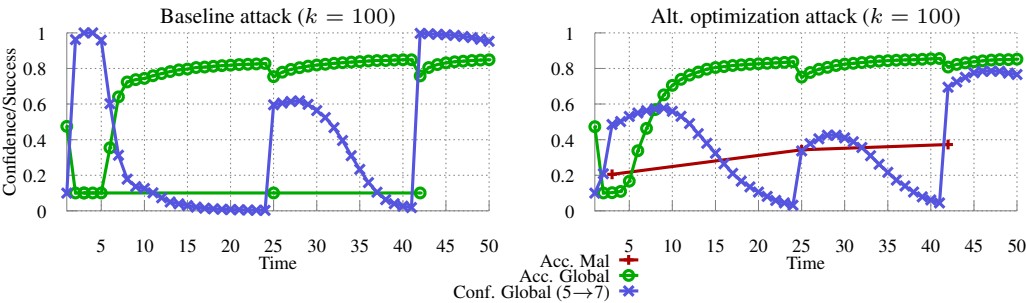

Figure 8: Metrics of interest for the baseline and alternating minization attack with $k = 100$ agents for the Fashion-MNIST dataset.

## A.2 RANDOMIZED AGENT SELECTION

When the number of agents increases to $k = 100$, the malicious agent is not selected in every step. Further, the size of $|\mathcal{D}_m|$ decreases, which makes the benign training step in the alternating minimization attack more challenging. The challenges posed in this setting are reflected in Figure 8, where although the baseline attack is able to introduce a targeted backdoor, it cannot ensure it for every step due to steps where only benign agents provide updates. The alternating minimization attack is also able to introduce the backdoor, as well as increase the classification accuracy of the malicious model on test data. However, the improvement in performance is limited by the paucity of data for the malicious agent. It is an open question if data augmentation could help improve this accuracy.

## B VISUALIZATION OF WEIGHT UPDATE DISTRIBUTIONS

Figure B shows the evolution of weight update distributions for the 4 different attack strategies on the CNN trained on the Faishon MNIST dataset. Time slices of this evolution were shown in the main text of the paper. The baseline and concatenated training attacks lead to weight update distributions that differ widely for benign and malicious agents. The alternating minimization attack without distance constraints reduces this qualitative difference somewhat but the closest weight update distributions are obtained with the alternating minimization attack with distance constraints.

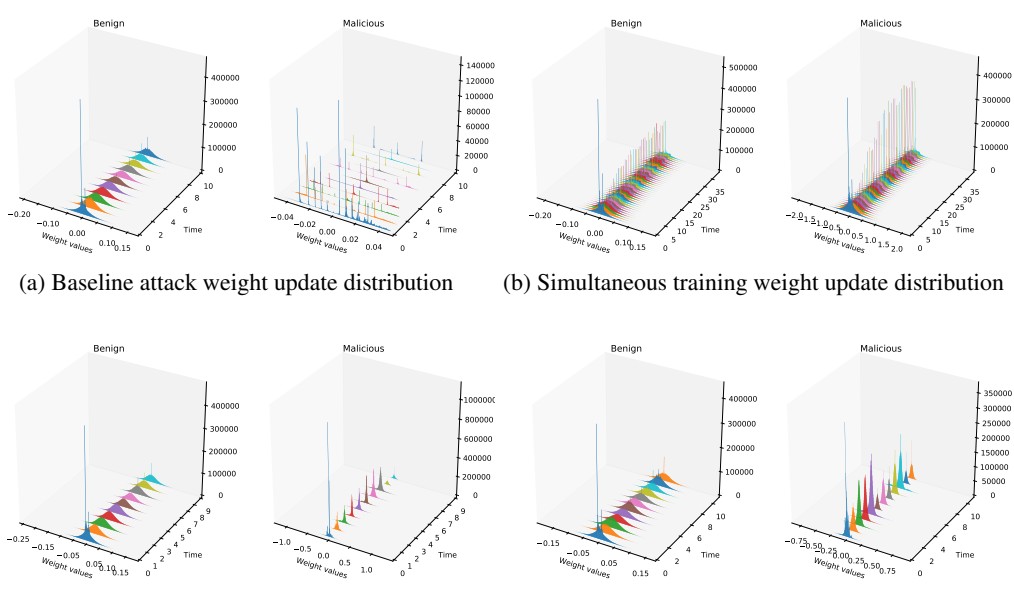

(a) Baseline attack weight update distribution  (b) Simultaneous training weight update distribution

(c) Alternating minimization attack weight update dis-(d) Alternating minimization attack with distance con-
tribution  straints weight update distribution

Figure 9: Weight update distribution evolution over time for all attacks on a CNN for the Fashion MNIST dataset.

## C  BYPASSING BYZANTINE-RESILIENT AGGREGATION MECHANISMS

Blanchard et al. (2017) recently proposed a gradient aggregation mechanism known as 'Krum' that is provably resilient to Byzantine adversaries. We choose to evaluate Krum as it is efficient, provably resilient and can be used a building block for better mechanisms Mhamdi et al. (2018). As stated in the introduction, the aim of Byzantine adversaries considered in this work and others (Chen et al. (2017b); Mhamdi et al. (2018); Chen et al. (2018); Yin et al. (2018)) is to ensure convergence to ineffective models. The goals of the adversary in this paper are to ensure convergence to effective models with targeted backdoors. This difference in objectives leads to 'Krum' being ineffective against our attacks.

We now briefly describe Krum. Given $n$ agents of which $f$ are Byzantine, Krum requires that $n \geq 2f + 3$. At any time step $t$, updates $(\boldsymbol{\delta}_1^t, \ldots, \boldsymbol{\delta}_n^t)$ are received at the server. For each $\boldsymbol{\delta}_i^t$, the $n - f - 2$ closest (in terms of $L_p$ norm) other updates are chosen to form a set $C_i$ and their distances added up to give a score $S(\boldsymbol{\delta}_i^t) = \sum_{\boldsymbol{\delta} \in C_i} \|\boldsymbol{\delta}_i^t - \boldsymbol{\delta}\|$. Krum then chooses $\boldsymbol{\delta}_{\textbf{krum}} = \boldsymbol{\delta}_i^t$ with the lowest score to add to $\mathbf{w}_i^t$ to give $\mathbf{w}_i^{t+1} = \mathbf{w}_i^t + \boldsymbol{\delta}_{\textbf{krum}}$.

In Figure 10, we see the effect of our attack strategies on Krum with a boosting factor of $\lambda = 2$ for a federated learning setup with 10 agents. Since there is no need to overcome the constant scaling factor $\alpha_m$, the attacks can use a much smaller boosting factor $\lambda$ to ensure the global model has the targeted backdoor. Even with the baseline attack, the malicious agent's update is the one chosen by Krum for 34 of 40 time steps but the global model is unable to attain high test accuracy. The alternating minimization attack ensures that the global model maintains relatively high test accuracy while the malicious agent is chosen for 26 of 40 time steps. These results conclusively demonstrate the effectiveness of model poisoning attacks against Krum.

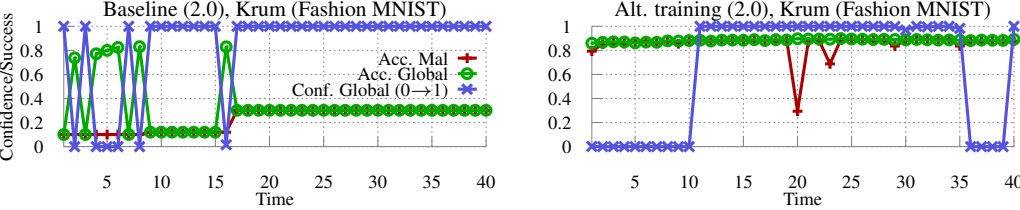

Figure 10: Metrics of interest for 2 different attack strategies with the Krum aggregation mechanism.

