# OpenReview forum: "Analyzing Federated Learning through an Adversarial Lens"
_ICLR.cc/2019/Conference_

### Official Review · AnonReviewer2 · 2018-10-29
**Proposing a new setting, interesting for ICLR and has some proof-of-concept results**

**Rating:** 6
**Confidence:** 4

**Review:**

The paper proposes a novel adversarial attack on deep neural networks. It departs from the mainstream literature in two points:
1. A 'federated' learning setting is considered, meaning that we optimize a DNN in parallel (imagine a map-reduce approach, where each node performs SGD and then a central server (synchronously) updates the global parameters by averaging over the results of the nodes) and an attacker has control over one of the nodes.
2. The treat model is not the common data poisoning setting, but 'model poisoning' (the attacker can send an arbitrary parameter vector back to the server).

The paper, which is well written, starts with proposing a couple of straightforward (naive) attacks, which are subsequently used as a baseline. Since there (apparently) is no direct related work, these baselines are used in the experimental comparisons. Then the authors propose a more sophisticated attacks, based on alternatingly taking a step into the attack direction (to get an effective attack) and minimizing the loss (to Camouflage the attack), respectively. They add also the feature of restricting the solution being not to far away from the usual benign SGD step.

All in all, I am acknowledging that his paper introduces the federated learning paradigm to 'adversarial examples' subcommunity of ICLR and would make for good discussions at a potential poster. I find the used method slightly oversimplistic, but this is maybe fine for a proof of concept paper.

Final judgement: For me this paper is a 6-7 rating paper; a nice addition to the program, but not a must-have.

A have a question to the authors that is important to me: it seems that the baseline attack could be very very simply detected by checking on the server the norm of the update vector of the attacked node. Since the vector has been boosted, the norm will be large. While your distance-based regularization somewhat takes that effect away, it remains unclear to what amount. Can you give me some (empirical) details on this issue? / or clarify if I am completely off here?  thank you

---

> ### Author Response · Authors · 2018-11-27
> **Details on norm-based detection**
>
> We thank the reviewer for their thoughtful comments. Indeed, detection using norm-based approaches is possible and is something we address explicitly in the paper. In Figure 3 (Page 6), we show the spread in the $L_2$ distances between the update vectors of the different agents. For the baseline attack, the spread of the distances of the malicious update vector from the benign updates diverges from that of the benign updates from each other over time.
>
> On the other hand, for the alternating minimization attack with and without distance constraints, the spread of distances for the malicious update vector approaches that of the benign vectors from each other as model training converges. This does not rule out detection but is a significant improvement over the baseline.

---

### Official Review · AnonReviewer1 · 2018-11-05
**The paper adresses model poisoning, a situation where it is relatively easy (and extremely important) to formally prove the claims (e.g. prove guaranteed convergence), yet not a single formal guarantee is provided.**

**Rating:** 4
**Confidence:** 5

**Review:**

The paper considers the federated learning setting as introduced by McMahan et al. (2017) and aims at securing it against model poisoning attacks.

Cons:

While I appreciated the writing clarity of the paper, the paper misses the whole point of defensive ML research: in the model poisoning case, a minimal requirement for a defense mechanism is to be formally proven *whatever is the behavior of the attacker* (within the threat model). Experiments alone are not sufficient for this purpose given the size of the space of possible attacks. Especially that (unlike evasion attacks) proofs are relatively easy to be made in the poisoning case.

For instance the literature cited by the paper (Chen 2017, Chen 2018, Blanchard 2017) + the recent follow-ups ((1)Alistarh et al. NIPS 2018, (2) El Mhamdi  et al. ICML 2018,  (3) Yin et al. ICML 2018 etc) are full of approaches the authors can follow to formally support their claims.
Also, the literature review has been done very lightly: Chen et al. 2017b (And most cited above) do *not* assume a single Byzantine agent as said in the paper, but assume up to <50% malicious (potentially colluding) agents.

Besides absence of formal support, how does the approach compare to the optimal results in (1) and (3) at least in the convex case ?
In the abstract, it is said (ii) that in the i.i.d situation, it will be easy to make spurious update standout among benign ones), this was proven wrong in (2) when the dimension of the model is large and the loss function highly non-convex, the case of neural networks for example. As a general comment, the defense mechanisms of the paper are all relying on a distance computation and thus will all provide the sqrt(d) leeway for an attacker as described in (2) and will fail preventing high-dimensionality attacks.


Pros:

I was very excited by the ideas in section 5, this work is the first to my knowledge to attempt at interpreting poisoning attacks. I suggest to the authors to either fix the issues mentioned above (and formally analyze their work), or to focus more on the interoperability question, if they want to keep the paper in the empiricist nature.

---

> ### Author Response · Authors · 2018-11-27
> **Clarifying that the intent of the paper is to demonstrate model poisoning attacks on neural networks in the federated learning setting, and that the detection mechanisms are used to generate more sophisticated attacks**
>
> We thank the reviewer for their insightful comments. We provide details for the aspects the reviewer found unclear and have correspondingly updated the paper.
>
> ‘...defensive ML research...’: We would like to clarify that our paper focuses on offensive ML research. The question we ask in this paper is: given that a particular agent in the federated learning setting is malicious, what specific behavior can it induce in the global model? One possible induced behavior is to prevent the convergence of the global model, which is the focus of attacks introduced in papers such as Blanchard et al. However, we deem the possibility of the introduction of a targeted backdoor, where the global model misclassifies just one or a few examples to be more interesting from an attack perspective and that is the focus of our paper. We show that, in the federated learning setting, an attacker with the ability to poison just a single model can cause a specific example to be misclassified with 100% confidence while ensuring that the global model achieves high accuracy on the test set (Figures 4a and 7).
>
> ‘Experiments alone are not sufficient...’: The purpose of the detection techniques introduced in the paper is to ensure that our attacks cleared what we deemed to be the minimum bar for stealth. Our intent is not to put forth these detection techniques as ways to fully secure a distributed learning system but to design attacks even under considerable restrictions on the adversary. This approach aided us in the development of more sophisticated attacks that are able to simultaneously insert targeted backdoors and meet minimum levels of attack stealth by bypassing basic detection schemes.
>
> ‘...the literature cited by the paper…’: We appreciate the list of previous papers the reviewer has referred to and would like to point out a few important differences. None of the papers referred to analyze the exact behavior the adversary can induce at the global model and are concerned mainly with behavior in the presence of arbitrary gradients. In our paper, we show that an adversary can induce the global model to provide incorrect outputs for a few examples while classifying the rest correctly, which is very different from an adversary preventing global convergence. In fact, in Appendix C, we show that the Byzantine-resilient aggregation mechanism Krum (Blanchard et al.) is not robust to our attack. The Krum defense chooses a single agent at each time step depending on a score derived from distance functions, instead of linearly aggregating a number of agents’ updates. Our attacks are stealthy enough that the score function of the malicious agent’s update causes it to be selected in a majority of time steps. We apologize for our misstatement regarding Chen et al. and have corrected this in the updated version.
>
> Further, the attacker model used in previous work assumes that the adversary has visibility into model updates from the benign agents even at the current time step. We do not make this assumption as it strikes us as unrealistic from an attack perspective, once again highlighting the fact that our paper seeks to understand attacker capabilities under a variety of conditions.
>
> ‘...relying on a distance computation...’: We note that one of our detection mechanisms, namely accuracy checking, does not rely on distance computations and is qualitatively different from the Byzantine resilient aggregation mechanisms proposed in previous work. Our baseline attack is thus easily detectable by the accuracy checking mechanism (Figure 1a) and Section 3.2.1), implying that at least some high dimensionality attacks can be detected. Our second detection mechanism does rely on distance computations, however, it is also able to detect updates sent using the baseline attack as aberrant (Figure 3). We introduce the alternating minimization attack precisely to overcome this detectability.
>
> ‘...absence of formal support...’: Following McMahan et al., our work focuses on analyzing possible attacks on neural networks and is thus of an empirical nature. While we appreciate the importance of formal results and have begun work on formalizing the attacks presented in this paper, we would like to emphasize that we choose to attack neural network based systems as a first step due to their widespread adoption. We wished to demonstrate the feasibility of our attack on state-of-the-art systems to begin the exploration of model poisoning attacks. Our attacks do however provide a lower bound on attacker success but do not rule out the possibility of more powerful ones. In the updated version of the paper, we have also added experiments with another qualitatively different dataset in order to further corroborate our claims.

---

### Official Review · AnonReviewer3 · 2018-11-06
**Interesting line of work but need quite some clarifications**

**Rating:** 5
**Confidence:** 4

**Review:**

This paper presents an interesting adversarial strategy to attack federated learning systems, and discussed options to detect and prevent the attacks. It is based not upon data poisoning attacks, but model poisoning attacks. It analyzes different strategies on the attacker's side, discusses the effect with real experimental data, and proposes ways to prevent such attacks from the federated learning perspective.

It is an interesting line of work which develops specific optimization algorithms to try to manipulate the global classifier for certain desired outcomes. I particularly appreciate the authors' thought process of improving the attack strategies with the understanding of the detection strategies. Also the authors proposed visualization to interpret poisoned models. However, I feel this paper needs major revision to make it a solid piece of work:
- Need better motivations. Is there any benefit to exploit model poisoning as opposed to data poisoning? Which one is more effective in attacking (and therefore harder to detect)?
- It's confusing to read through Section 3 on these different attack strategies. For instance, in 3.2 the authors introduced explicit boosting and implicit boosting, but only explicit boosting is focused because implicit boosting didn't show good results in Figure 2. But is there a setup that implicit boosting will be beneficial (to the attackers)? I feel the authors introduced many strategies, but didn't give theoretical analysis. It is hard to pick the "best" attack strategy in practice, thus making it equally hard to have the "best" detection strategy.
- The figures are also confusing in that it's hard to understand what the 3D figures are trying to show, and it is not obvious what the legend means. The authors should also explain whether this experimental observation is unique to this data set/experimental setup or has similar trends in similar federated learning settings.
- Clearly Appendix A is unfinished

I encourage the authors to address these questions carefully and resubmit the manuscript later.

---

> ### Author Response · Authors · 2018-11-27
> **Clarifying motivation, attack strategies, figures and experimental setup**
>
> We thank the reviewer for their nice suggestions for clarifying the paper and have updated the paper accordingly. The Appendix now contains more results for a different dataset, more agents and a different aggregation mechanism.
>
> ‘...benefit to exploit model poisoning...’: In the federated learning setup, due to privacy concerns, the agents do not share their data directly with the server. Each agent performs computation on its local data and sends a model parameter update to the server. This is the reason we focus on model poisoning attacks since an attacker has the ability to directly poison the model update to achieve their goals. While the poisoning could be done by modifying the local data, the scope of possible changes induced in the model by data poisoning is subsumed by those possible with model poisoning. It is also unlikely that the server can rule out certain updates on the basis of them being inconsistent with the agent’s local data since it has no visibility into that data. Thus, we believe at least in the federated learning setup, our focus on model poisoning attacks is pertinent and well-justified.
>
> ‘...introduced many strategies...’: Implicit boosting is much more resource hungry as compared to explicit boosting and adds a large (5x) overhead in terms of the time taken for the malicious agent to generate weight updates. In a practical setting, this can lead to the malicious agent being dropped due to synchronization considerations. It is also much less effective (Figure 2). Since we have chosen to experiment with neural networks, following McMahan et al., our attacks do not come with provable guarantees. However, we believe that we have taken an important first step in demonstrating the feasibility of introducing stealthy backdoors in neural networks during the process of federated learning using model poisoning and our attacks can serve to provide lower bounds on attacker success. Further, since our attacks are able to achieve 100% confident targeted misclassification on multiple datasets, provable attacks will only be able to improve in terms of efficiency and stealth.
>
> ‘figures are also confusing’: We apologize for the lack of clarity in the figures (Figures 1b), 1c), 4b) and 4c)) depicting the model update distributions. We have replaced them with a representative time slice which we hope depicts the differences between benign and malicious updates more clearly.
>
> ‘unique to this data set’: In Appendix B.1., we have added results on the Adult Census data to demonstrate that our observations are not restricted to just a single dataset. The Adult Census data is qualitatively different from Fashion-MNIST, yet the same general attack observations hold. Our attacks are able to achieve high confidence misclassification while ensuring that the global model achieves high accuracy on the test set.

---

### Author Response · Authors · 2018-11-27
**A revised version of the paper has been uploaded**

We have uploaded a revised version of the paper. In particular, the revised version contains the following changes:

1. The ‘related work’ paragraph has been updated to include the papers suggested by Reviewer 1 and to better place our work with respect to those papers.
2. The Experimental Setup (Section 2.3) now contains details of the UCI Adult Dataset which is the second dataset we use to show the effectiveness of our attacks.
3. Figures 1 and 4 have been updated to show time slices of the weight update distributions which we believe more clearly demonstrate our attack insights. We thank Reviewer 3 for the suggestion to improve the figures. The 3-dimensional time evolution plots of the weight update distributions are now in Appendix B.
4. Figure 3 has been updated with a different color scheme to make it easier to read. Its caption has also been updated for clarity.
5. Appendix A.1. has been added to demonstrate results for our attacks on the Adult Census dataset. This provides evidence for the effectiveness of our attack across datasets.
6. Appendix A.2. now discusses the case when the malicious agent is chosen at random (due to the presence of a large number of agents) and its implication for attack success.
7. Appendix C contains results demonstrating the effectiveness of our attack against Krum, a Byzantine resilient aggregation mechanism.
8. The abstract has been shortened and the introduction updated to reflect the changes above.

A number of other minor writing and presentation changes have also been made to improve the flow of the paper. We welcome further comments!

---

### Meta-Review · Area_Chair1 · 2018-12-16
**interesting model but could benefit from better writing and comparisons**

**Confidence:** 3
**Recommendation:** Reject

**Metareview:**

This paper proposes model poisoning (poisoned parameter updates in a federated setting) in contrast to data poisoning (poisoned training data). It proposes an attack method and compares to baselines that are also proposed in the paper (there are no external baselines). While model poisoning is indeed an interesting direction to consider, I agree with reviewer concerns that the relation to data poisoning is not clearly addressed. In particular, any data poisoning attack could be used as a model poisoning attack (just provide whatever updates would be induced by the poisoned data), so there is no good excuse to not compare to the existing strong data poisoning attacks. One reviewer raised concerns about lack of theoretical guarantees but I do not agree with these concerns (the authors correctly point out in the rebuttal that this is not necessary for an attack-focused paper). I do feel there is room to improve the overall clarity/motivation (for instance, equation (1) is presented without any explanation and it is still not clear to me why this is the right formulation).